# Making Cognitive Ergonomics in the Human–Computer Interaction of Manufacturing Execution Systems Assessable: Experimental and Validation Approaches to Closing Research Gaps

Andreas Dörner [1], Marek Bures [1,*] , Michal Simon [1] and Gerald Pirkl [2]

1   Mechanical Engineering, University of West Bohemia, Univerzitny 2732/8, 301 00 Pilsen, Czech Republic; a.doerner@oth-aw.de (A.D.); simon@fst.zcu.cz (M.S.)
2   Electrical Engineering, Media and Computer Science, OTH Amberg-Weiden, Kaiser Wilhem Ring 23, 92224 Amberg, Germany; g.pirkl@oth-aw.de
*   Correspondence: buresm@fst.zcu.cz

**Abstract:** Cognitive ergonomics and the mental health of production workers have attracted increasing interest in industrial companies. However, there is still not much research available as it is regarding physical ergonomics and muscular load. This paper designs an experiment to analyze the cognitive ergonomics and mental stress of shop floor production workers interacting with different user interfaces of a Manufacturing Execution System (MES) that is adjustable for analyzing the influence of other assistive systems, too. This approach is going to be designed with the Design of Experiments (DoE) method. Therefore, the respective goals and factors are going to be determined. The environment will be the laboratories of the University of Applied Sciences Amberg-Weiden and its Campus for Digitalization in Amberg. In detail, there will be a sample assembly process from the automotive supplier industry for demonstration purposes. At this laboratory, the MES software from the European benchmark SAP is installed, and the respective standard Production Operator Desk is going to be used with slight adaptions. In order to make the cognitive ergonomics measurable, different approaches are going to be used. For instance, body temperature, heart rate and skin conductance as well as subjective methods of self-assessment are planned. The result of this paper is a ready-to-run experiment with sample data for each classification of participants. Further, possible limitations and adjustments are going to be discussed. Finally, an approach to validating the expected results is going to be shown and future intentions are going to be discussed.

**Keywords:** cognitive ergonomics; mental workload; human–computer interaction; manufacturing execution system; design of experiments; ergonomic assessment

## 1. Introduction

Manufacturing Execution Systems (MES) are a commonly used technology in industrial companies. Their principle is to link shop floor production with resource planning software as well as production scheduling in real time [1]. MES is usually implemented with an information and communication technology in the respective work center [1,2]. For instance, touch-screen desktops with a graphical user interface (GUI) are a pretty common solution. The European benchmark for industrial software SAP is indeed providing standard templates for the interface using touch screens [2].

In modern Industry 4.0 environments, the shop floor workers in production are already confronted with lots of information and communication with different kinds of data, systems, colleagues, etc. These systems, as well as other future technologies such as augmented and virtual reality, smart devices, voice control and gesture control, may overwhelm some workers. This might have an influence on the respective cognitive

ergonomics [3]. Whether information overload can have an influence on mental stress has not been researched sufficiently. While physical workload and stress have been researched a lot in the past, their cognitive load and its influence are still not fully transparent [4].

A previous study shows that the result for a search string on the database Scopus for physical workload in correlation with assembly and manufacturing has almost twice as many hits as the respective string screening for mental workload [4]. Table 1 demonstrates this statement, adjusted with current figures from January 2024.

**Table 1.** Results for a first exemplary search on the platform Scopus (adapted from [4]).

| Wording for Search in Title-Abstract-Keywords | Scopus Results 2024 |
| --- | --- |
| (physical AND workload AND assembly OR manufacturing) | 436 |
| (mental AND workload AND assembly OR manufacturing) | 263 |

The necessity of this topic was noticed in Dörner, Bures and Pirkl's (2022) latest publication which underlines the need for further research in cognitive ergonomics and mental workload in an Industry 4.0 environment [4]. Their results highlight respective research gaps in mental workload, especially with Industry 4.0 technologies or assistive technologies in general. Only three papers are included in the synthesis of their review [3–6].

Therefore, this article designs an experiment to analyze the cognitive ergonomics and mental stress of shop floor production workers interacting with different user interfaces of a Manufacturing Execution System that remains adjustable for taking other assistive systems into consideration. The result should help close the research gap shown in [4].

## 2. Materials and Methods

### 2.1. Design of Experiments

Design of Experiments (DoE) is an approach for changing smaller characteristics of an existing process and trying to analyze the effects of these changes on different key results [7–9]. A production process in the form of a simple illustration is shown in Figure 1 [10]. The input and output run through the process itself with influencing input factors. During DoE, input factors are changed and the effect on the output is analyzed (statistically or non-statistically). The output can have an effect on the (semi-finished) part itself, but also key indicators of industrial engineering such as quality figures, scrap rate, efficiency and—in the case at hand—human factors and ergonomics [7–9].

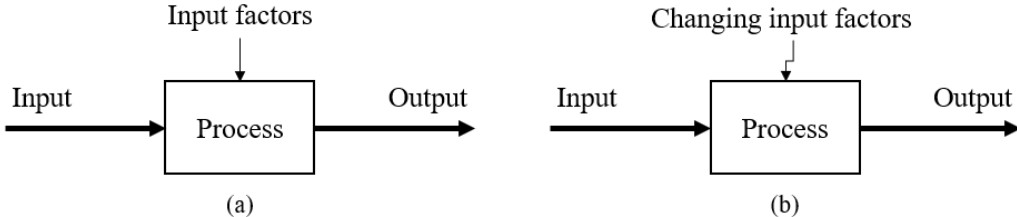

**Figure 1.** General model of a manufacturing process: (**a**) in original setup; (**b**) in experimental setup (adapted from [10]).

DoE aims to plan an industrial experiment 'in an optimal way with single or multiple underlying objectives, e.g., cost minimization, effective resource consumption, reduced environment pollution, etc.' [11]. Therefore, the Design of Experiments method is commonly used for optimization purposes. As resources in industrial companies are usually limited, more experiments than necessary are avoided. One of the major concerns in DoE, independently from the main objective, has always been the limited number of experiments [11].

The DoE method usually has four major steps, starting with determining goals (measuring cognitive ergonomics), followed by choosing factors (user interface design), adjust-

ing the level (level of information load from low over medium to high) and evaluating results [7–9,11].

### 2.2. Measuring Cognitive Ergonomics

As already mentioned, the purpose of this paper is to define an experiment capable of measuring the cognitive ergonomics and mental workload of production workers when the user interface of the MES system is changed. These changes should be related to the information load on the screen; for instance, different scenarios with different information loads on the GUIs should be designed.

In order to make cognitive ergonomics and mental stress measurable, different techniques have been found to be useful. On the one hand, there are objective methods for physiological measurement [12,13]. For instance, heart rate, the galvanic skin response (GSR) and body temperature have been both statistically relevant for measuring mental workload and suitable for application in production [12,13]. On the other hand, subjective methods, such as self-assessment of the participant, are used too. There are several standardized questionnaires, such as the NASA Task Load Index (TLX) and many more [14,15].

For the experiment at hand, wearable sensors for the GSR and heart rate will be used. Further, a stationary thermal camera will be used to track the participants' body temperature. The adjusted NASA TLX questionnaire is shown in Figure 2 [14,15]. The participants will complete the questionnaire after they have gone through the experiment and rate the six questions from very low to very high, respectively, from 0 to 20 points, as the marked clustering shows.

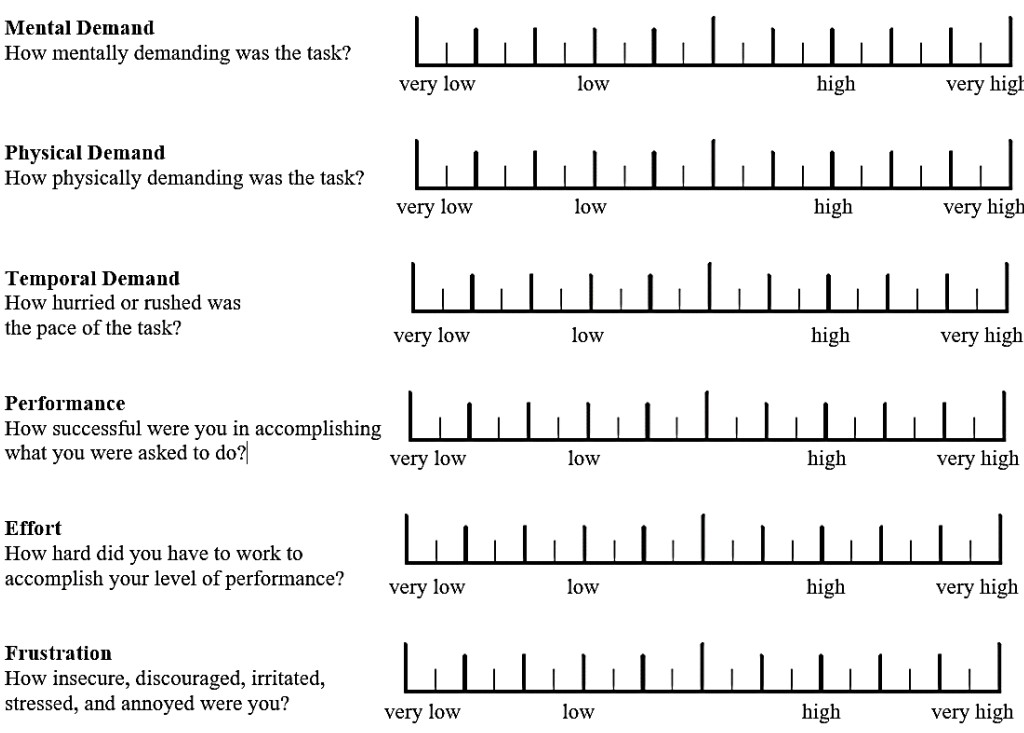

**Figure 2.** Adjusted NASA TLX Survey [14,15].

### 2.3. Manufacturing Execution Systems

Manufacturing Execution Systems interact with shop floor workers through a user interface, offering various possible solutions. One traditional approach involves a desktop interface, where workers initiate and confirm production orders individually using a screen or a mouse as input hardware. Contemporary methods include voice control, gesture control or chatbots for data input during production. Another prevalent option is a touch monitor positioned at the work center, commonly supplied by software companies specializing in Enterprise Resource Programs (ERP) or MES software. These technologies are

assistive technologies, and still have a research gap regarding their influence on cognitive ergonomics according to the review of Dörner, Pirkl and Bures (2022) [4].

The user interface, a crucial component encompassing both software and hardware, dictates how individuals interact with the device and understand the displayed information. Effective user interfaces provide comfort and convenience, facilitating access to all information within the software to achieve its purpose. The user interface we deal with in the experimental design is the standard GUI of the German software company SAP and its template for touch screens. However, there should be the possibility to analyze assistive technologies other than touch screens with a similar experiment. Therefore, the goal of this article is to create a universal experiment that allows us to close the research gap in cognitive ergonomics.

## 3. Results

### 3.1. Experimental Design

#### 3.1.1. Work Center and Process Design

For the experiment itself, there will be three different test environments. All three will have the same exemplary assembly process. It is a simplified process that is adapted from a similar work center of an industrial company in the automotive supply industry. In a nutshell, the process consists of two components that are assembled. The demonstrator is located at the Digital Campus in Amberg. Figure 3 shows the shop floor illustration of the work center. It is a work center for the assembly of simple products in order to test different assistive systems as well as information and communication technology (ICT). In this case, the ICT we want to evaluate is the Production Operator Dashboard (POD) of a Manufacturing Execution System that is visualized as a touch screen on one side of the work center. At this work center, the MES software SAP is installed. The work center itself is right in front of the shop floor worker, with the material supply on the side. Smaller modifications to this layout are no problem.

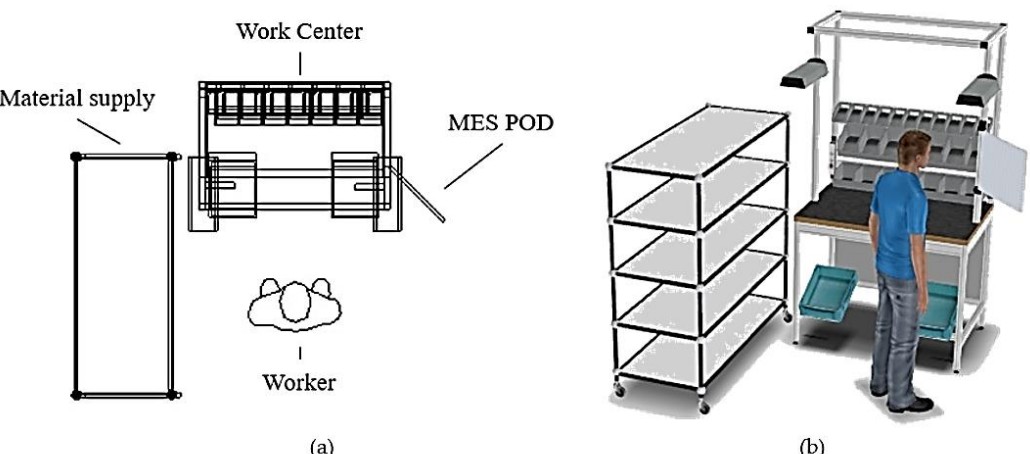

**Figure 3.** (**a**) Shop floor layout of the work center; (**b**) 3D illustration.

Even though cognitive ergonomics and mental workload should be analyzed rather than physical workload in this experiment, a 3D illustration helps in visualizing the overall ergonomical situation in the laboratory. Figure 3b shows such an overview. It shows that the workstation is trivially designed, laid out for a greater variety of assembly processes, whereby both semi-finished and finished goods can be fed in. Smaller bulk material can be stored at the desk and communication can take place via the touch-screen monitor.

The process in this exemplary experiment is as follows. A rubber seal is added to a frame using screws to merge it with the respective bracket. This rubber seal should decrease noise and vibration when the product is used by the customer.

On this work center, no further fixtures are in use, just a tool for screwing. The semi-finished parts are in a trapezoidal form, as is the rubber seal. The parts are available in

different variants, differing in their surface and material. For instance, black ones, carbon ones and aluminum ones are available. Table 2 lists the steps of the routing and the respective work instructions.

**Table 2.** Routing and work instruction for sample assembly process.

| Nr. | Routing | Work Instruction |
|:---:|:---:|:---:|
| 010 | Start production order in ME | Start |
| 020 | Take semi-finished part and rubber seal from top shelf of material supply | Semi-finished part   Rubber seal |
| 030 | Grasp rubber seal and needed number of screws | |
| 040 | Assemble rubber seal on semi-finished parts using the given tool | |
| 050 | Place finished part on lower shelf of material supply | Complete |
| 060 | Post production order in ME | |

With this information in hand, the 3D illustration can be described in more detail. The screws and tools are stored as bulked material at the work center, and the semi-finished parts as well as the components (rubber seal) are delivered on the left side of the material supply (Figure 4). After the assembly process, finished parts are stored on the lowest shelf of the material supply. The experiment will be recorded using cameras that are mounted on the work center to make every cognitive reaction transparent and comprehensible.

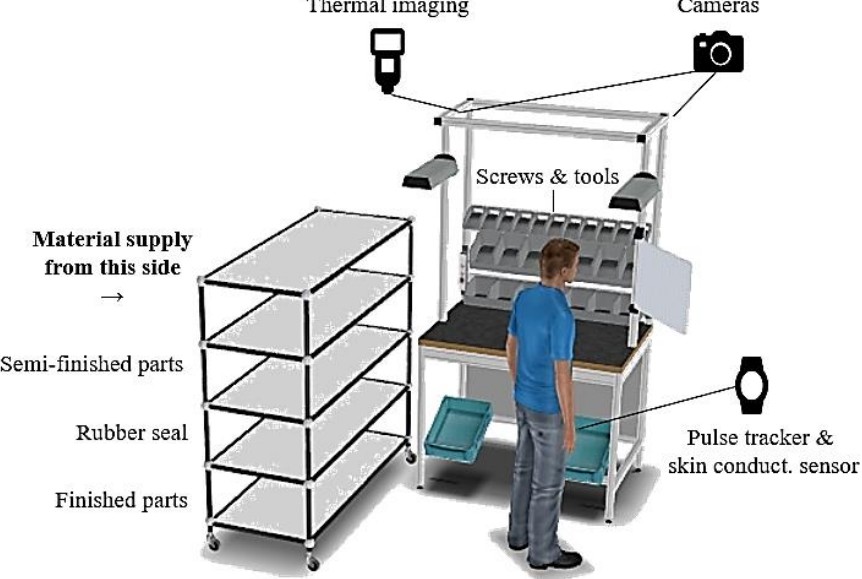

**Figure 4.** Three-dimensional illustration of the work center for the experiment including information on work instruction and measurement.

As already mentioned, body temperature, heart rate and skin conductivity as well as subjective evaluation are commonly used approaches to make cognitive ergonomics measurable [12,13]. The first three possibilities are going to be measured during the experiment by using thermal imaging for body temperature measurement mounted to the work center, as well as pulse trackers and skin conductivity sensors linked to the participant. A subjective evaluation will take place after the experiment for each participant using the NASA TLX scale.

### 3.1.2. User Interfaces in Use

In the first scenario, the SAP standard POD is going to be used. Commonly used for MES in production, this standard POD is also provided as a solution for touch-screen monitors [1]. Figure 5 depicts a very rudimentary example for a fictional production process of bracket assembly. It shows basic user information ('PEGGY' and 'HILL'), work center information like the abbreviated name 'BASSY573', a work list of seven production orders as well as different standardized color-coding symbols and icons for different features. For instance, core functionalities, like starting and completing an order, obtaining further information on work instructions or data acquisition for the assembled parts, are already linked in this standard template. Also, other abbreviations like shop floor control (SFC), data collection (DC) and quantity (Qty) are mentioned.

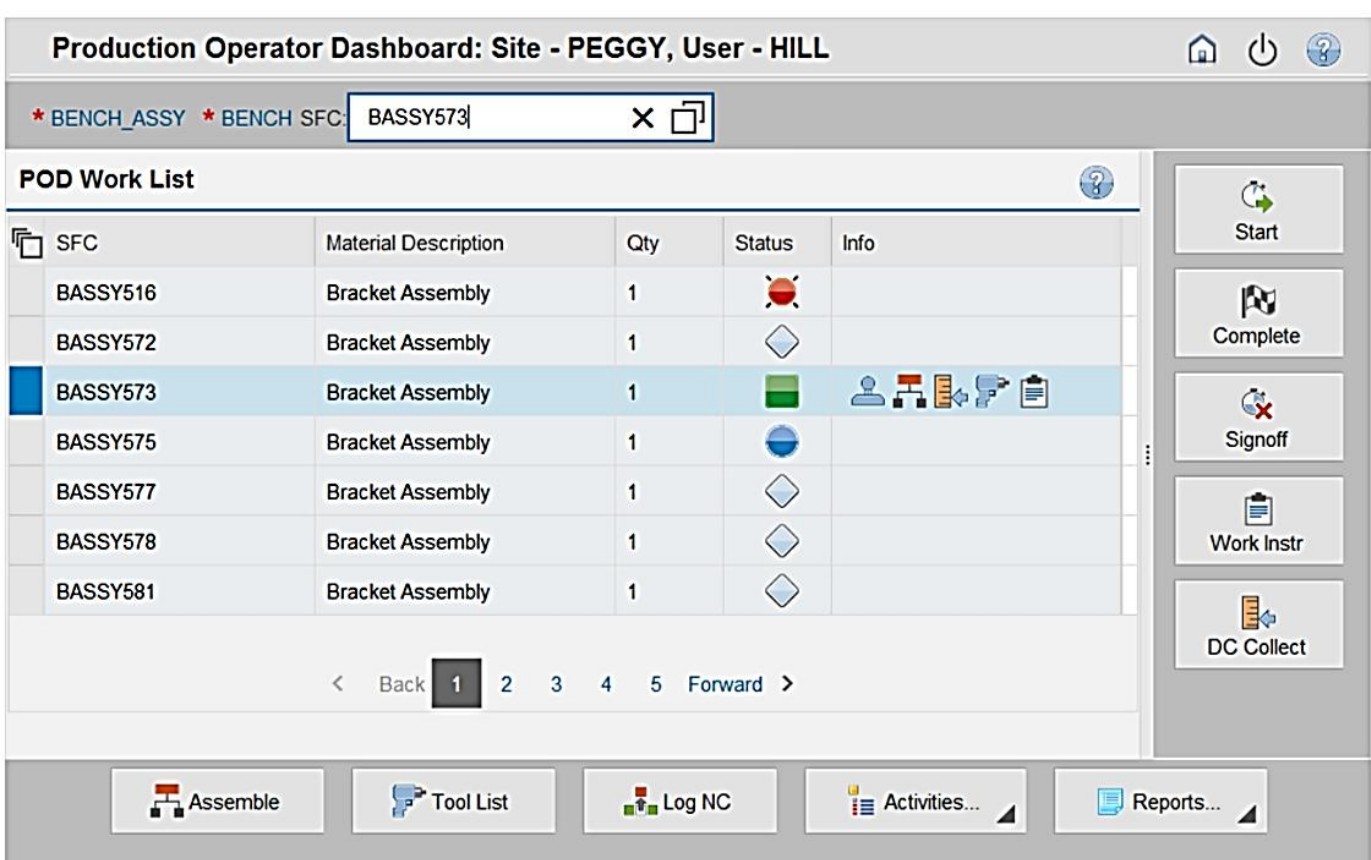

**Figure 5.** MES Standard POD with highest information load for Scenario 1 [2]. © Copyright 2024. SAP SE. All rights reserved.

The second scenario will have a user interface that has already become leaner. The POD work list has been shortened to roughly one shift of several hours and only the five most used icons are visualized on the main screen.

Coming to the third scenario with an even leaner user interface, all further added, but not necessary, information for the worker to execute the assembly steps has been removed.

Also, additional information in the form of icons and the current production orders have been left out. Thus, the icons for personal information and logout on the upper right corner have been summarized to one icon with a menu function.

As is typical for the DoE approach, only a few things are changed to make them compare well with the other scenarios. In this case, the three scenarios work with the same shop floor layout, routings, parts and work instruction, but with a different user interface of the MES.

So, the participants would run the experiment for a certain amount of time in one of the three scenarios (see Table 3). The chronology of the scenarios may vary and be different for each participant to eliminate possible correlations. Also, whether one participant runs through one or all alternatives may vary, too.

**Table 3.** Summary of the three test scenarios.

| Test Scenario | Scenario 1 | Scenario 2 | Scenario 3 |
|---|---|---|---|
| **Information Load** | Highest | Medium | Low |

The experimental design is quite simple on purpose. There should be the possibility to execute this experiment in different locations. For instance, some industrial companies from the university's partner circle use the same Manufacturing Execution System as the laboratories at OTH. Therefore, there is an opportunity to build this experiment (e.g., with other products) at industrial companies as well, to obtain further data from experienced production shop floor workers.

Figure 6 summarizes the three test scenarios visually. It underlines the changes from the GUI with the highest (a), medium (b) and lowest (c) information load. In the laboratory in Amberg, the SAP standard POD (a) were converted into these two leaner versions using the information gained from the eye-tracking of sample users. After validating some samples, we will discuss whether these layouts are suitable for the planned experiment or if they are too similar to create a statistically relevant difference.

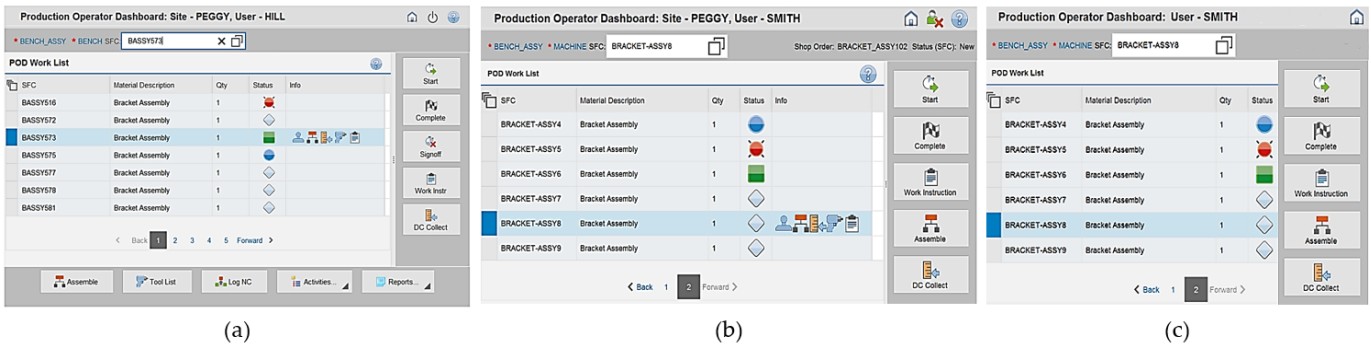

(a)  (b)  (c)

**Figure 6.** Overview of the three test scenarios: highest (**a**); medium (**b**); and lowest (**c**) information load [2]. © Copyright 2024. SAP SE. All rights reserved.

### 3.1.3. Participants Planned

The aim is to achieve a similar number of participants to similar research projects. Wu et al. (2016) and Ustunel and Gunduz (2017) performed their experiment with around 40 participants, and Gueltieri et al. (2022) had only 14 (see Table 4) [3–6].

**Table 4.** Summary of similar studies [3–6].

| Author | Year | Title | Number of Participants |
|---|---|---|---|
| Wu et al. [3] | 2016 | Influence of information overload on operator's user experience of human-machine interface in LED manufacturing systems | 38 |
| Ustunel and Gunduz [5] | 2017 | Human-robot collaboration on an assembly work with extended cognition approach | 40 |
| Gualtieri et al. [6] | 2022 | Evaluation of Variables of Cognitive Ergonomics in Industrial Human-Robot Collaborative Assembly Systems | 14 |

The result of this review is an average of 32 attendees, which is therefore the goal of this experiment. The participants are divided into age groups and experience levels. This makes it possible to analyze further whether the influence differs between highly experienced assembly workers and people who have never worked in industrial production before. Furthermore, the age group can give additional data that can be used for researching the demographic trend and its influence in using modern information and communication technologies in industrial production. The age groups are divided into a younger and an older one. The average age of industrial workers in Germany has been analyzed a lot in the recent years and also has been the subject of many forecasts in the past decade [16–21]. Therefore, a cumulated age of 45 years will be used for limiting the age groups of assembly workers in industrial production. Ranasinghe et al. (2023) [22] as well as Clark and Ritter (2020) [23] mentioned that when most members of a certain group are older than 45, the group has to deal with an aging workforce. Overall, the cluster for the participants is shown in Figure 7. It is divided into four quarters.

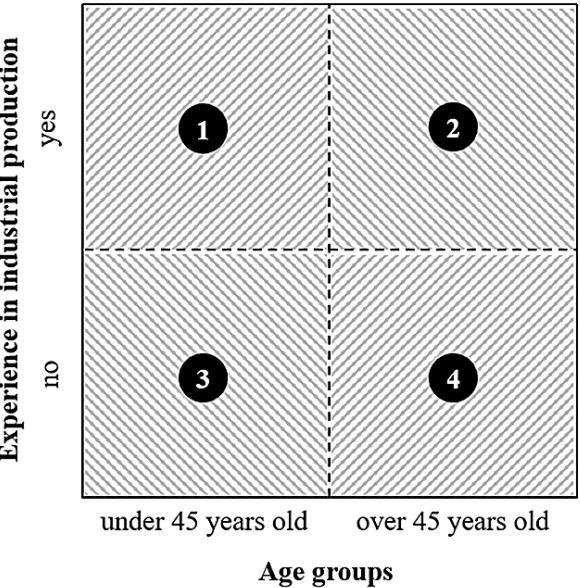

**Figure 7.** Cluster for the participants according to their age and experience in industrial production.

*3.2. Sample Data from Pilot Study*

For all of the four quarters shown, a sample of the experiment is taken in order to evaluate it. This should give the possibility to perform certain adjustments (see Section 4) before executing the experiment with the intended 32 (or more) participants. The attendees run through scenario two of the experiment, meaning they have worked with a user interface with a medium information load. In this sample, all of them assembled a total of seven production orders in the assembly process shown in Table 2. An overview of the participants is summarized in Table 5.

**Table 5.** Overview of the sample participants.

| Number (Figure 7) | Socio-Demographic Data | Experience in Industrial Production | Scenario (Table 3) |
|---|---|---|---|
| 1 | Age: 27 Gender: male | yes | Scenario 2: medium information load |
| 2 | Age: 65 Gender: male | yes | Scenario 2: medium information load |
| 3 | Age: 28 Gender: female | no | Scenario 2: medium information load |
| 4 | Age: 63 Gender: female | no | Scenario 2: medium information load |

All four participants need roughly 15 min to execute the task of the experiment, which includes completing seven production orders in the assembly process shown in Table 2. Every production order consists of one semi-finished part (lot-size 1).

Table 6 depicts the first indicator measured in a sample of four participants. It shows the average heart rate of each participant as well as the minimum and maximum values.

**Table 6.** Overview of measured heart rate during the sample assembly process.

| Number (Figure 7) | Heart Rate Min | Heart Rate Max | Average Heart Rate |
|---|---|---|---|
| 1 | 71 beats/min | 83 beats/min | 74 beats/min |
| 2 | 79 beats/min | 89 beats/min | 83 beats/min |
| 3 | 72 beats/min | 85 beats/min | 74 beats/min |
| 4 | 84 beats/min | 90 beats/min | 86 beats/min |

The heart rate was measured without any handicap for the participants. This yielded valid results, but all of the figures are in the lowest area (below 115 beats/min) according to the used pulse tracker.

The measurement of body temperature, however, had its limitations. As shown in Figure 8, different locations of the thermal imaging camera were tried out prior to the sample pilot study. The limitation is visualized below: The camera focusses on a certain point. But in an assembly work station, the shop floor workers, and especially their foreheads, do not remain in the same position over the whole time of the process. After these try-outs, it was decided to retain the camera position shown in Figure 4, since this was closest to the participant and their head. This location results in the thermal imaging in Figure 8c, showing 31.2 °C.

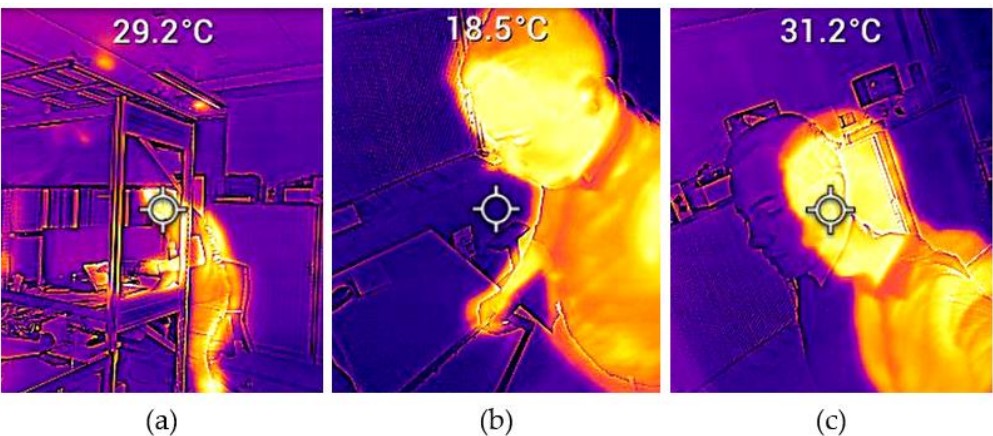

(a)      (b)      (c)

**Figure 8.** Results of different try-outs of the position for the body temperature camera: (**a**) next to the work center; (**b**) above the participant; (**c**) next to the participant.

With this location, the four sample participants went through the experiment. Figure 9 illustrates the course of the body temperature of participant 1 over the time of the experiment. It clearly shows that the camera focus is not targeting the participant's body over the complete time of the experiment. This is evident from the number of drops, when the measured thermal image drops from over 30 °C (focusing on the worker' skin) to less than 15 °C (focusing on inventory of the laboratory) within one second. Nevertheless, after the try-outs, the final position of the camera is still able to show the current situation of the participants whenever its focus is on their body. This brings the possibility to interpret temperature peaks, as the graph in Figure 9 shows. Therefore, body temperature might not be able to serve as the only indicator for cognitive ergonomics in an industrial production environment, but there is still the possibility to recognize situations of higher workload due to heat rushes in the worker. In the social sciences, these situations are called psychotic fever and are an indicator of a higher mental load. The temperature of the fever can vary. For some participants it is 37–38 °C, and for others, it is even 40 °C or more [24]. However, the resulting graphs from this experiment can at least depict when body temperature is rising quickly over a short period of time. Due to this, thermal imaging can underline cognitive ergonomics in this experiment, even though it is not suitable to serve as the only measurement.

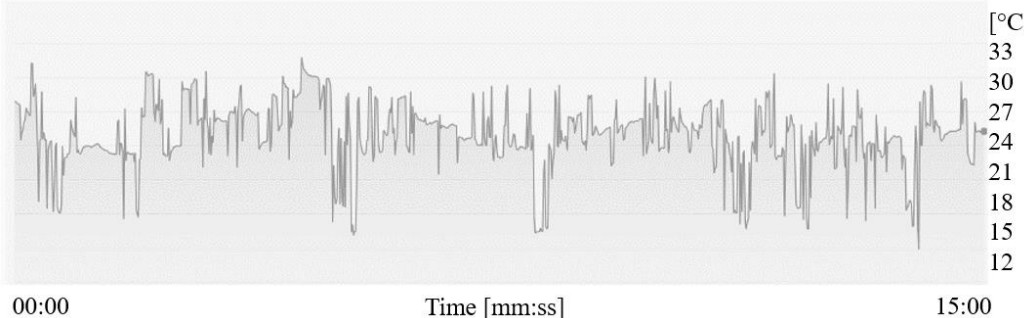

**Figure 9.** Body temperature of sample participants over the time of the experiment.

The galvanic skin response is usually measured in the unit of Siemens (S). Figure 10 exemplarily demonstrates the measurement of participant 1. The overall values of the GSR are quite low in comparison to other studies that use similar sensors. The highest value of all four sample participants in the pilot study is 0.15 µS. The graphs of all four participants look quite similar, with the lowest values at the start of the experiment of around 0.015 µS and the highest value in participant 1 of 0.15 µS. Thus, besides the minimum and maximum values, the graph itself is also similar for all participants. Overall, all GSR measures can be clustered into three phases. The first phase, phase A, starts for all subjects at around 0.015 to 0.02 µS and increases quite steadily for the first 5–10 min of the experiment. This can be seen as a side effect of the sensor itself, as the finger tapes can cause sweating between the sensor and tape, especially during the first minutes of wearing. Furthermore, there is phase B, in which the peaks of the GSR measurement of all participants are detected. In this phase, the participants have the most interaction with the user interface. Last but not least, there is phase C. This remains fairly constant below the maximum value. As already mentioned, the maximum value of 0.15 µS for participant 1 is not as high as it is in other studies. For instance, Nourbakhsh et al. (2017) measures GSR rates of 2–3 µS when their participants were stressed [25]. So, the maximum is almost 10 to 20 times higher than in this pilot study. A possible adjustment would be to make the three test scenarios more extreme and less similar compared to those suggested in this paper. A further adjustment and discussion can be found in the Discussion.

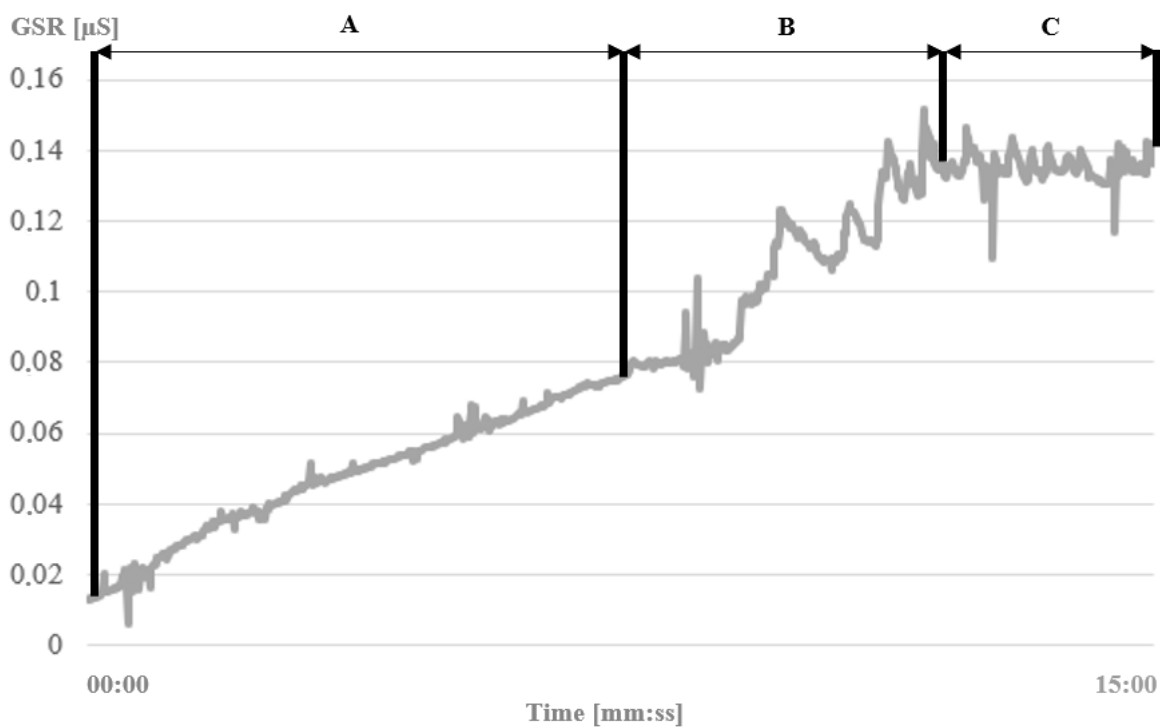

**Figure 10.** Skin conductance of sample participants in the experimental work center.

The last indicator for the cognitive ergonomics of the participants suggested in this pilot study is the subjective rating in a NASA TLX survey. Table 7 summarizes the results and the respective average of the answers given by the four sample participants. All four delivered valid answers in each of the six items. Besides the mental demand, which has an average score of 8.00, it can be recognized that performance was rated quite high by the participants; effort, on the other hand, was rated quite low. This fits the expectations. This sample process seems fit for purpose since it is not the main challenge for the worker, and a clear result can be achieved when changing the user interfaces. The standard deviation ranges between 2.16 and 3.74 for all six items. Possible answers for the evaluation would have been from 1 to 20 for each category.

**Table 7.** Summary of the NASA-TLX rating of the four sample participants.

| NASA-TLX Item | Number of Valid Results | Average | Standard Deviation |
|---|---|---|---|
| Mental demand | 4/4 | 8.00 | 2.16 |
| Physical demand | 4/4 | 4.75 | 3.11 |
| Temporal demand | 4/4 | 4.50 | 2.65 |
| Performance | 4/4 | 11.25 | 2.65 |
| Effort | 4/4 | 3.75 | 2.75 |
| Frustration | 4/4 | 5.25 | 3.74 |

These participants and their measurements for cognitive ergonomics give a first hint for the scale and order of magnitude that can be expected after the experiment. It also helps to discuss and support the consideration of necessary adjustments, which are summarized in Section 4.

### 3.3. Validation Method

Ergonomics in general can be evaluated by making them assessable and measurable. However, to validate the overall ergonomics of a work center, different variables need to be considered. For example, the Ergonomics Assessment Work Sheet is a commonly used

approach to make physical ergonomics and muscular load visible, without the need to look at every single indicator [26]. It summarizes the overall risk situation on a scale, with color-coded areas. Figure 11 shows the risk scales from the green area, over the yellow one, to the red area [26].

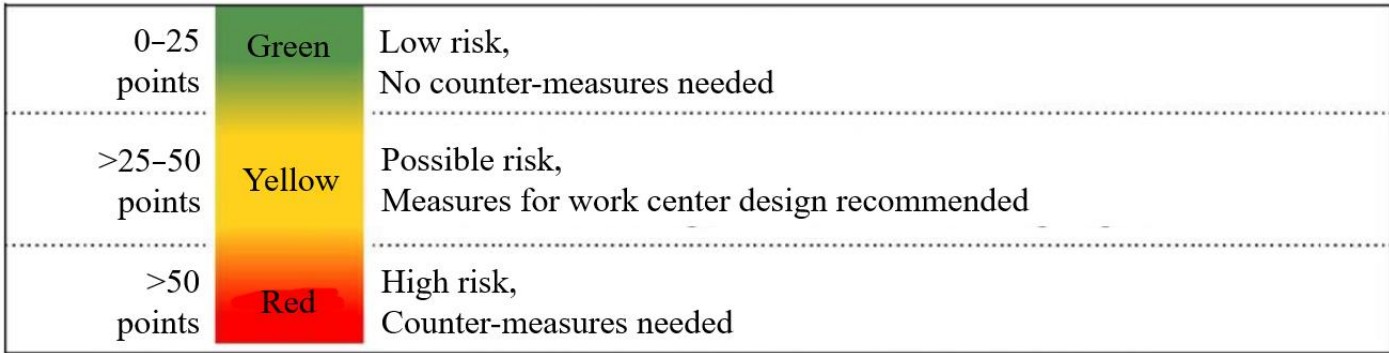

**Figure 11.** Ergonomic validation example from EAWS [26].

Similar to this, a color-coded overview of the overall work center's cognitive ergonomics could be given. As already mentioned by other authors, the heart rate and the GSR figures provide decent results for mental load [12,13]. The sample measurements underline this. They also show that the delta of these two measurements is more suitable than analyzing the absolute figures due to the participants' individual bodies. Body temperature will be used as an add on to analyze heat rushes and psychological fever. In Figure 12a, the delta of the participants' heart rate as well as the galvanic skin response is visualized. When both are quite high, it indicates that the mental load in this case is higher than in cases with both values being relatively low. If subjective measurement is included, the validation could look like the color-coded cube in Figure 12b.

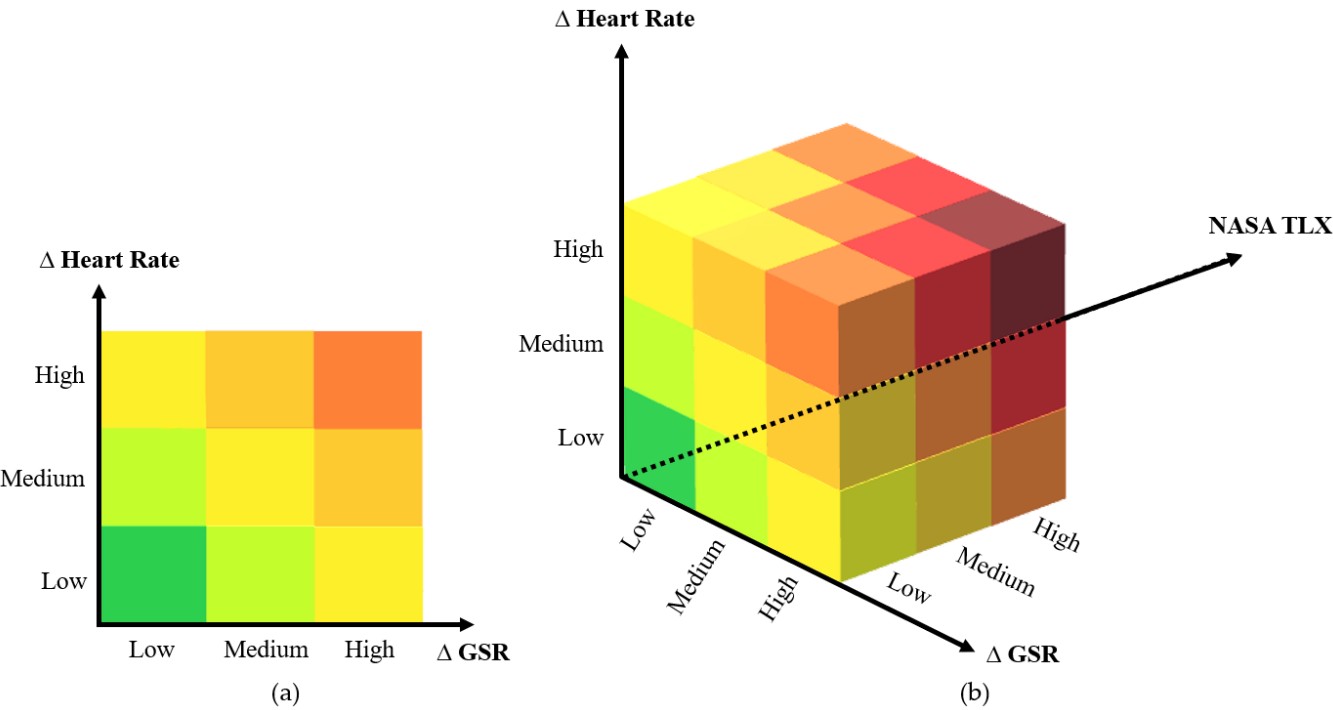

**Figure 12.** Validation approach for cognitive ergonomics for rating the results of the experiment: (**a**) with physiological measurement; (**b**) with physiological measurement and subjective evaluation.

After obtaining the results of more experiments, future research ideas consist of creating numbers behind the different colors. For now, there are colored blocks for each low, medium and high value. In this schematic approach, the scale consists of points from 1 to 9 (a) or even from 1 to 27 (b) when the subjective figures from the task load index are included (see Figure 13).

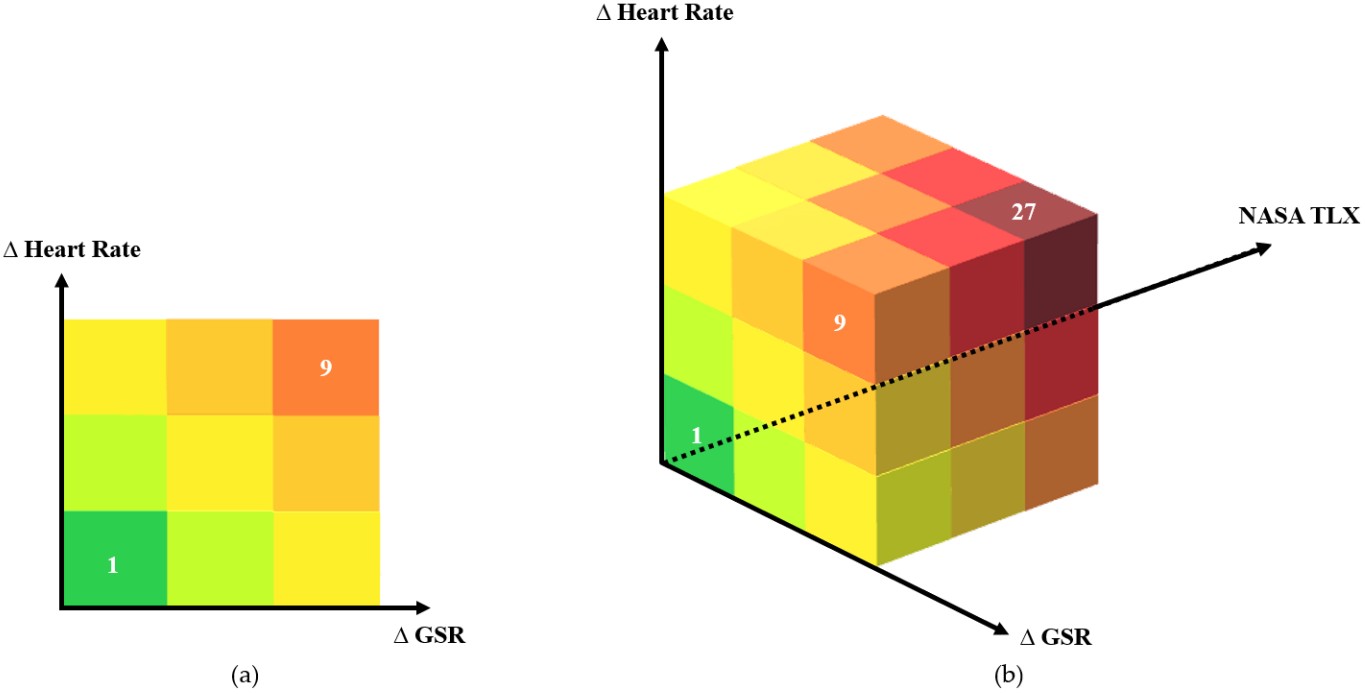

**Figure 13.** Validation approach for cognitive ergonomics including a schematic scale: (**a**) with physiological measurement; (**b**) with physiological measurement and subjective evaluation.

The actual scale could be adjusted after executing the experiments, as explained in this article, with different assistive systems, adjustments and participants.

## 4. Discussion

All in all, there should be many opportunities to apply this experiment to different types of assistive technologies, and then, analyze them. Hence, in this particular case, it was supposed to analyze the influence of the mental workload before and after the optimization of a graphical user interface of a piece of MES software with lean management methods. Besides the standard MES POD and an optimized, lean one, whether it makes sense to add another GUI to the test environment could be discussed. For example, an even more information-heavy user interface could be added to the test or the one with the lowest information load could be replaced to check the results. For the upcoming experiment, it is also an option to increase the differences between these three user interfaces, to make the influence on cognitive load more visible. For the experiment itself, it might also make sense to create mock-ups of GUIs that are more different, because the measurements of the indicators galvanic skin response and heart rate were quite small. It is possible that the participants' reactions will be more significant if the three scenarios differ more from each other. Therefore, there would be a mock-up similar to the one with medium information load that was used in the pilot study, but the ones with more and less information load would differ much more from the medium scenario.

Another idea was to analyze the participants' age in relation to the results on their cognitive ergonomics, especially in Europe, where the aging workforce has become a major topic among Industry 4.0 and Industry 5.0 trends [16,21]. A possible outcome that can lead to future research is that cognitive ergonomics could either be more or less affected when shop floor workers are dealing with assistive technologies. Not only touch

screens, but also other information and communication technologies like eye-tracking, voice and gesture control, and virtual and augmented reality, can lead to synergies or target conflicts with the demographic trend in Europe. If there are meaningful differences in the results of this experiment depending on the age of the participant, this could be a future research intention.

Besides the execution of this experiment in the laboratories in Amberg, it is also possible to expand the experiment to different locations. The campus in Weiden, the laboratories at the university of West Bohemia in Pilsen as well as different industrial companies outside of the university's partner circle are conceivable ideas. It is also possible to change the respective assembly process and evaluate the results with other products, different material supplies or different assistive technologies. If cooperation with an industrial company and their production workers as participants works out, it is also possible to differ between a novice group (e.g., less than one year of experience) and an experienced group (e.g., more than three years of experience) of participants, just like Wu et al. (2016) did in their experiment [3]. In the current setup, the differentiation is between no experience and some experience in industrial production.

However, the current design also has its limitations. For instance, the thermal imaging can be improved in order to receive consistent results for the participant's body temperature. Also, the NASA-TLX approach is state-of-the-art, but has the limitation of subjective evaluation. In order to compensate for these disadvantages of these measurements, our validation approach focusses on skin conductance and heart rate since they have high statistical relevance according to Arkouli et al. (2022) [12].

## 5. Conclusions

This article provides a standardized experimental design that is easily adjustable for different technologies and processes. It should help in creating various experiments to assess cognitive load and close the respective research gap. Table 8 summarizes the idea of this paper and the outcome of the three most similar studies found in Dörner, Pirkl and Bures' (2022) literature review [4]. These are the same findings that are mentioned in Table 4. In addition, the experiment planned in this thesis is presented in Table 8, which summarizes the classification of participants, the method and the hypotheses to be evaluated.

**Table 8.** Summary of the experimental design in comparison to similar studies [3–6].

| Paper | Participants | Method | Results |
|---|---|---|---|
| Wu et al. (2016) [3]: Influence of information overload on operator's user experience of human-machine interface in LED manufacturing systems | Total of 38 participants 21 male, 17 female Glasses, no glasses and lenses Novice group as well as experts | Three prototypes of sorting system for LED production with a user interface with low, medium and high complexity | • Lower complexity decreases user's attention and higher complexity significantly increases their attention. Novice participants feel significantly higher levels of effort, frustration and mental workload. <br> • Information overload increases cognitive workload and decreases user efficiency |
| Ustunel and Gunduz (2017) [5]: Human-robot collaboration on an assembly work with extended cognition approach | Total of 40 participants 22 male, 18 female | Four different groups, two for each gender and also two with and without extended cognition approach | • No significant difference in cognitive load between genders. Findings showed that designing workplaces with the approach of extended cognition could reduce cognitive load on workers |

**Table 8.** *Cont.*

| Paper | Participants | Method | Results |
|---|---|---|---|
| Gueltieri et al. (2022) [6]: Evaluation of Variables of Cognitive Ergonomics in Industrial Human-Robot Collaborative Assembly Systems | Total of 14 participants with no previous experience with collaborative robots and minimal experience in performing assembly activities | Three different scenarios with changing features and interaction modalities including low interaction (1), compromised (2) and compromised with added speed modification (3) | • Trust increased and frustration decreased with the enhancement of workstation features and interaction conditions by shifting between the various scenarios<br>• Cognitive workload increased slightly in medium compared to low interaction, while it decreased considerably in Scenario 3 compared to Scenario 2 |
| Experiment planned in this article | At least 32 participants with and without experience in industrial production, as well as a group above and a group under 45 years old | Participants are going to go through a sample assembly process, interacting with one (or more) user interfaces. The GUIs will differ in their information load, from low to medium to high | The following hypotheses are going to be evaluated<br>• The higher the information load on the MES user interface, the higher the mental workload of the production worker becomes<br>• The influence of information load on workers' cognitive ergonomics differ depending on their age |

The different setups might support in creating a more comprehensive validation approach and can give first data to make the ranking more detailed than is shown in this paper. Therefore, more participants in the current layout as well as more experimental setups are planned as future research intentions.

Also, the age groups of the participants will help in conducting further research on the influence of assistive technologies on the aging workforce. This will become more relevant considering the demographic trends industrial companies in many countries are facing [16,22].

Finally, this experiment, with possible minor adjustments, is suitable for other, similar research projects, especially on assistive systems other than robots. Two out of the three findings have their focus especially on human–robot collaboration. However, there are slight differences that help close the research gaps mentioned in the Introduction [4].

**Author Contributions:** Conceptualization, A.D. and M.B.; methodology, A.D. and M.B.; software, A.D.; validation, A.D.; formal analysis, A.D.; investigation, A.D.; resources, M.B., M.S. and G.P.; data curation, A.D.; writing—original draft preparation, A.D.; writing—review and editing, M.S., M.B. and G.P.; visualization, A.D.; supervision, M.S.; project administration, M.S. and G.P.; funding acquisition, A.D., M.B. and G.P. All authors have read and agreed to the published version of the manuscript.

**Funding:** This research was funded by BTHA, Bavarian-Czech Academic Projects, grant number JC-2022-28T and this paper was created with the subsidy of the project SGS-2024-032 entitled "Intelligent production system" granted by the Internal Grant Agency of the University of West Bohemia in Pilsen.

**Data Availability Statement:** The study data can be made available upon request to the corresponding author.

**Conflicts of Interest:** The authors declare no conflicts of interest.

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
