# Peer review of "Making Cognitive Ergonomics in the Human–Computer Interaction of Manufacturing Execution Systems Assessable: Experimental and Validation Approaches to Closing Research Gaps"

_machines, doi:10.3390/machines12030195_

Round 1
Reviewer 1 Report
Comments and Suggestions for Authors
Dear authors,
The article presents an approach to experimentation and validation aimed at addressing research gaps in the field, focusing on the mental health and cognitive ergonomics of production workers. The study aims to analyze the influence of different user interfaces of a Manufacturing Execution System (MES) on the cognitive ergonomics and mental stress of production floor workers. The approach was designed using the Design of Experiment (DoE) method. The article is very well-organized, informative, and engaging. I particularly appreciated the article's structure.
While the article presents an interesting and relevant approach to analyzing cognitive ergonomics in manufacturing execution systems, there are a few areas where improvements could be made: (1) the study could benefit from a larger sample size to enhance the validity of the results; (2) the paper could provide more information about the study's limitations and potential future research directions; (3) it would be helpful to include more details about the statistical methodology used in data analysis; (4) the first paragraph of the conclusion should be incorporated into the discussion; (5) the conclusion does not adequately support the study and should be revised.
However, I do not believe that these suggestions diminish the merit of the research, and I recommend approval after addressing the improvements mentioned in items 2 to 5.
I hope these suggestions are helpful.
Author Response
(1 and 2) We added further potential research intentions, also with a higher number of participants, and elaborated limitations during the discussion/conclusion; (3) We described the statistical evaluation more detailed. Therefore, we added e.g. the standard deviation if the task load index; (4) We rearranged some parts of discussion/conclusion, also addressing ideas from other reviewers; (5) We pointed out, how the table in the conclusion is used to support the study and to close research gaps while linking the table with the introduction.
Reviewer 2 Report
Comments and Suggestions for Authors
Thank you for the opportunity to review paper entitled: Making Cognitive Ergonomics in Human-Computer Interaction of Manufacturing Execution Systems assessable: Experiment and Validation Approach to close Research Gaps.
This paper presents a timely and significant contribution to the field of cognitive ergonomics, particularly focusing on the mental health of production workers in industrial settings. The rising importance of cognitive ergonomics in contrast to the more extensively researched area of physical ergonomics and muscular load is a critical point of discussion in contemporary industrial studies. This paper is a good attempt to improve production system via ergonomics.
The core of the paper is the design of an experiment analysing the cognitive ergonomics and mental stress of shop floor production workers, particularly in their interaction with different user interfaces of a Manufacturing Execution System (MES). The novelty of this study lies in its adaptability for analysing the influence of various assistive systems, broadening its applicability beyond the immediate scope of the MES.
Employing the Design of Experiment (DoE) method, the paper meticulously outlines the process of defining specific goals and factors for the study. The choice of the University of Applied Sciences Amberg-Weiden and its Campus for Digitalization in Amberg as the environment for this experiment adds to the credibility and relevance of the research, given its focus on a sample assembly process from the automotive supplier industry.
The integration of the MES software from SAP and the use of the standard Production Operator Desk, albeit with slight adaptations, is a strategic choice that aligns well with the paper's objective. It provides a real-world context to the experiment, enhancing the study's practical applicability.
One of the paper's strengths is its comprehensive approach to measuring cognitive ergonomics. This is not often in scientific literature. The use of both objective measures like body temperature, heart rate, and skin conductance, and subjective self-assessment methods gives a holistic understanding of the workers' cognitive ergonomic state (well-being). This dual approach is crucial in capturing the complex nature of cognitive ergonomics and mental stress.
Nevertheless, the ergonomic nature of the presented research also has its drawbacks, as the methods used were largely prepared for production purposes and are characterised by simplistic and rather uncomplicated characterisation of, for example, NASA TLX, of which I am personally a fan.
The considerations presented are an interesting way of combining many different ergonomic dimensions. Nevertheless, one can get quite a few imperfections in it, which hinder a positive reception of the presented content to a rather large extent. Among these imperfections, let me point out the following:
why the table number one of the results of searches for specific phrases in the literature had to be adapted from another publication with data from over a year ago. I did a very quick search and it was all too easy to update this table by the way it could be shown that there is quite a large increase in interest in this topic. Over the past year there have been more than 70 publications in the first query.
The drawings presented in the article are of rather low quality, both in terms of graphic aesthetics and partly in terms of the factual contribution they make - this is the case, for example, with drawing number one, which shows a rather poor and uninspiring manufacturing model. The drawings that follow, such as a view of a section of a NASA map, also have little to do with science. The following drawings are supposed to show fragments of an experiment, but you can't get the impression that they are taken from someone's thesis. The views shown in the following figures 5, 6, 7 and 8 do not add much to the work either, especially as the quality is very poor.
From my point of view, it is interesting to see the attempts to present different types of evolution of workloads using different scenarios, be it video camera telemetry measurements or body temperature, although it would be useful to present in a more systematic way some kind of summary of these methods in terms of their applicability and to evaluate them on the one hand biblio-metrically and on the other, perhaps, the potential for conducting similar experiments in the future. There also seems to be a lack of critical commentary on the individual methods because, for example, the galvanic skin response seems to be a very insensitive measure for various types of phenomena occurring at the operator's position. Figure eleven, on the other hand, shows the change in temperature, with a range of 15° to 30°. This would require some comment, as it is difficult to make any reference to this value, since even the skin temperature should not vary so much, not to mention the various mechanisms by which the skin temperature normalises to the internal body temperature.
On the other hand, the summary of mean responses in Table 7 does not add much to the study either. The NASA TLX method itself is a highly subjective method, and especially with a small number of responses, the value of the mean of the responses in each dimension does not add much. It is also difficult to comment positively on the 2 columns that show the same number of correct and incorrect answers in their entirety. A much better solution would be to present in these columns, for example, the standard deviation or a second value representing the share of a given factor in the sum of all ratings according to the NASA methodology.
At the end of the article, the concept of validating different types of tools is presented by comparing them with the 3 zones found in the EAWS method. This approach may of course be interesting, but to present it in a scientific article without carrying out an analysis is ornamental and it is difficult to conclude whether a method of evaluating fields based on classification of heart rate would work at the level of NASA ratings and galvanic skin response.
In view of these considerations and doubts, it is difficult to take a clear position on the article presented. On the one hand it is interesting and gathers a lot of different threads, but on the other hand it fails to bring them together coherently enough. Moreover, the coherence is disturbed by certain inconsistencies in the article presented and, finally, by the appearance of a table summarising the literature review on experimental design in the conclusions section. At this point, it would be appropriate to conclude the analysis carried out and it seems that the authors have decided that the best way to summarise the content presented is to introduce a review, probably from another source.
Overall, this paper could be a significant addition to the limited research on cognitive ergonomics in industrial settings. It could open new possibilities for improving the mental health and work efficiency of production workers through better understanding and designing of cognitive ergonomic environments. But within current state t is difficult for accomplishing this goal.
Author Response
(1) We updated the introduction and looked for respective research gaps. Also, this helps to draw a line to the table in the conclusion; (2) For Figure 1, we adjusted the cited one for a better fit in our article. Further, we cut some of the figures 5-8 since they have been obsolete. For the remaining ones, we improved the quality; (3) We discussed the methods/approach more critically in many places with the ideas from other reviewers; (4) We added further explanation for the range in the participant’s body temperature; (5) Table 7 has been changed. The standard deviation has been added and mentioned; (6) The idea of the validation approach has been added because the multiple future research intentions based on this experimental design might help to make it comprehensive. Also, it might be from potential interest for the readers of the special issue “Assessment, Validation and Improvement of Safety and Ergonomics in Human-Robot Interaction”; (7) As mentioned in (1), we explained research gaps that the table in the conclusion is referring to.

Reviewer 3 Report
Comments and Suggestions for Authors
Correct some spelling errors and verb tenses as suggested.
Question 1: The subject addressed in this article is worthy of investigation
4) Agree: the work is interesting and the research question is clear. The evaluation of the human factor in the field of collaborative robotics is a hot and very current topic in industry 4.0. Evaluating how different environmental factors influence performance in human cobot interaction is of research interest.
Question 2: The information presented is new
4) Agree: To the best of my knowledge, yes the topic addressed is new.
Question 3: The conclusions are supported by the data
5) Agree: an analysis of the duration of the tasks was carried out depending on environmental conditions, such as whether or not a barrier was raised. A percentage calculation of the duration before and after was carried out. Perhaps a statistical analysis of the durations and a before-after comparison test would have made the analysis more precise, but perhaps it is outside the scope of this work.
Question 4: The manuscript is appropriate for the journal
4) Agree: The paper is in line with journal aim and scope.
Question 5: Organization of the manuscript is appropriate
4) Agree: the manuscript is well structured. I understand that the description of complex procedures requires the use of acronyms. For a non-expert, this slows down the reading a little (maybe insert a short legend?). The research question is well defined and the work well structured and precise.
Question 6: Figures, tables and supplementary data are appropriate
4) Agree: the graphs used are extremely explanatory and effective in my opinion!
In my opinion the work is well done, organized and interesting!
Author Response
For the percentage evaluation before/after, we added the standard deviation of the task load index and mentioned it. Further, we avoided acronyms in many places and added an explanation for many abbreviations while making sure that the abbreviations are mentioned first.
Round 2
Reviewer 2 Report
Comments and Suggestions for Authors
OK
Author Response
Thank you for the input. The articles quality improved thanky to this.
